# Assessment of Conventional Solvent Extraction vs. Supercritical Fluid Extraction of Khella (*Ammi visnaga* L.) Furanochromones and Their Cytotoxicity

**DOI:** 10.3390/molecules26051290

**Published:** 2021-02-27

**Authors:** Noha Khalil, Mokhtar Bishr, Mohamed El-Degwy, Mohamed Abdelhady, Mohamed Amin, Osama Salama

**Affiliations:** 1Department of Pharmacognosy and Medicinal Plants, Faculty of Pharmaceutical Sciences and Pharmaceutical Industries, Future University in Egypt, Cairo 11835, Egypt; osalama@fue.edu.eg; 2Arab Company for Pharmaceuticals and Medicinal Plants, (Mepaco-Medifood), AlSharqiya 11361, Egypt; mbishr_2000@yahoo.com (M.B.); Degwys78@hotmail.com (M.E.-D.); magdi_science@hotmail.com (M.A.); alichem2005@yahoo.com (M.A.)

**Keywords:** *Ammi visnaga*, khellin, supercritical fluid extraction, visnagin, MCF-7

## Abstract

Background: Khella (*Ammi visnaga* Lam.) fruits (Apiaceae) are rich in furanochromones, mainly khellin and visnagin, and are thus incorporated in several pharmaceutical products used mainly for treatment of renal stones. Methods: The objective of this study was to compare the yield of khellin and visnagin obtained using different conventional solvents and supercritical fluid extraction (SCFE) with carbon dioxide (containing 5% methanol as co-solvent). Water, acetone and ethanol (30% and 95%) were selected as conventional solvents. Results: Highest extract yield was obtained from 30% ethanol (15.44%), while SCFE gave the lowest yield (4.50%). However, the percentage of furanochromones were highest in SCFE (30.1%), and lowest in boiling water extract (5.95%). HPLC analysis of conventional solvent extracts showed other coumarins that did not appear in supercritical fluid extraction chromatogram due to non-selectivity of solvent extraction. *Ammi visnaga* extracts as well as standard khellin and visnagin were tested for their cytotoxic activity using sulforhodamine B assay on breast cancer (MCF-7) and hepatocellular carcinoma (Hep G2) cell lines. Results revealed a strong cytotoxic activity (IC_50_ < 20 µg/mL) for the SCFE and standard compounds (khellin and visnagin) (IC_50_ ranging between 12.54 ± 0.57 and 17.53 ± 1.03 µg/mL). However, ethanol and acetone extracts had moderate cytotoxic activity (IC_50_ 20–90 µg/mL) and aqueous extract had a weak activity (IC_50_ > 90 µg/mL). Conclusions: Thus, supercritical fluid extraction is an efficient, relatively safe, and cheap technique that yielded a more selective purified extract with better cytotoxic activity.

## 1. Introduction

*Ammi visnaga* L. (Apiaceae) is a popular commercial medicinal plant indigenous to Egypt and widely cultivated in the Mediterranean region and Europe [1]. The plant is industrially important as it is employed worldwide in many pharmaceutical products; either its extract or isolated active constituents [2]. It is widely used for its antispasmodic properties [3]. It relieves kidney stones, used in some heart diseases like angina, as well as hypertriglyceridemia [4,5,6,7,8,9]. These medicinal effects are attributed to its coumarin content [10]. Two types of coumarins had been identified in *A. visnaga*: furanochromones (mainly khellin and visnagin) and pyranocoumarins (mainly samidin, visnadin and dihydrovisnadin) [11,12,13]. Recent studies also reported that khellin and visnagin, as well as their derivatives, are candidate drugs for treatment of several types of tumors, inflammatory diseases and epilepsy [14,15]. The amount of these compounds varies widely according to genetic as well as environmental factors [7,16]. Moreover, different extraction methods affect their yield. Individual compounds present in *A. visnaga* have different pharmacological activities. Khellin, for example is used to prevent renal crystal deposition [4,17], while visnagin is a strong vasorelaxant [18]. Therefore, there is a need for the individual separation of these compounds from other coumarins present in the plant. Several green techniques have been recently implicated for natural product extraction to achieve industrial sustainability. These include, for example, molecular imprinting technology, solvent-resistant nanofiltration, deep eutectic solvent-based microwave-assisted extraction and supercritical fluid extraction [19,20,21,22]. Supercritical fluid extraction (SCFE), especially with carbon dioxide, has been recently applied in different fields; including natural products, metal cation extraction, polymer synthesis and particle nucleation [23,24,25]. It has been applied in the extraction of several essential oils such as orange, lemon, hawthorn, chamomile, oregano and rose oils [26]. It has also been used for the extraction of carotenoids from *Spirulina platensis* strain Pacifica microalgae spirulina [27], tropane alkaloids from *Datura candida* [28], lycopene from tomatoes [29], azadirachtin from neem seed kernels [30], oleoresins from *Calendula officinalis* [31] and chromones from Saposhnikoviae radix [32]. Several studies have demonstrated that SCFE may be superior to different conventional extraction methods, with regards to cleanliness, selectivity, time and cost saving, as well as possibility of manipulating the composition of the extracts according to the desired use [33].

The present study aimed at comparing the yield of furanochromones (khellin and visnagin) from *A. visnaga* fruits using different solvents as well as SCFE technique, in addition to assessing the in-vitro cytotoxic activity of the obtained extracts. This is to provide an industrially applicable extraction method that is fast, selective, relatively safe and cost effective, as well as to evaluate the effect of extraction technique on the cytotoxic activity.

## 2. Results

### 2.1. Conventional Solvent Extraction vs. Supercritical Fluid Extraction of Furanochromones

Dry extract yield, total content of furanochromones, as well as their percentage yield in each prepared extract is shown in Table 1. Although *A. visnaga* aqueous extract was easily converted to dry powder, however, it gave the least yield of furanochromones (82.11 ± 2.1 g, 5.95% *w/w*). Ethanol 95% gave the highest yield of furanochromones amongst the tested conventional solvents (111.1 ± 2.4 g, 8.23% *w/w*). Although the 30% ethanol extract gave a relatively lower furanochromones yield (101.74 ± 1.9 g, 6.59% *w/w*), than that of ethanol 95%, its physical properties were much better, while also it is the most industrially economic and low-cost solvent used commercially worldwide. Supercritical fluid extraction of furanochromones from *A. visnaga* fruits yielded a white, slightly fatty extract with strong characteristic odor. This extract had the best yield of furanochromones (135.0 ± 52.34 g, 30.1% *w/w*).

### 2.2. HPLC Analysis of Conventional Solvents and SCF Extracts

HPLC fingerprint chromatogram of the SCF and conventional solvents extracts are shown in Figure 1. Khellin and visnagin retention times were 9.3 min. and 11.1 min., respectively. However, the HPLC chromatogram of conventional solvents extracts showed other minor furanochromones which did not appear in case of supercritical fluid extraction.

### 2.3. In-Vitro Cytotoxic Activity of A. visnaga Extracts, Standard Khellin and Visnagin

*Ammi visnaga* extracts, as well as standard khellin and visnagin, were tested for their cytotoxic activity using SRB assay. Results revealed a strong cytotoxic activity for the SCFE and standard compounds (khellin and visnagin) (IC_50_ ranging between 12.54 ± 0.57 and 17.53 ± 1.03 µg/mL); however, the ethanol and acetone extracts had moderate cytotoxic activity and aqueous extract had a weak activity (Table 2).

## 3. Discussion

*Ammi visnaga* L. is an important medicinally active plant which is incorporated in several pharmaceutical preparations used worldwide [2]. The main pharmacologically active compounds in this plant are the furanochromones; khellin and visnagin. Therefore, a convenient large-scale production technique is needed for those compounds. This study aimed at comparing the yield of furanochromones from *A. visnaga* fruits using different solvents, as well as SCFE technique, in addition to assessing the in-vitro cytotoxic activity of the prepared extracts. Boiling water, 30% ethanol, 95% ethanol and acetone were used as conventional solvents for extraction of furanochromones from *A. visnaga* fruits. Although water is available, safer, and cheaper than other solvents used, it showed many drawbacks such as need for long durations and much energy for its removal; also, furanochromones yield (82.11 ± 2.1 g, 5.95 % *w/w*) was lower than yields obtained in other methods. However, its extract was easily converted to dry extract. Ethanol 95% extract gave the highest yield of furanochromones (111.1 ± 2.4 g, 8.23% *w/w*), but its high cost is a major industrial drawback. Through different trials to replace the expensive ethanol by water and to reduce its hazards and cost, it was found that 30% ethanol gave the best extract and relatively lowered the cost with quite good yield and quality (101.74 ± 1.9 g, 6.59% *w/w*). Supercritical fluid extraction of furanochromones from *A. visnaga* fruits yielded a white, slightly fatty extract with strong characteristic odor. This extract had the best yield (135.0 ± 3.22 gm, 30.1% *w/w*) and best physical properties. HPLC chromatogram of all conventional extracts showed other minor compounds which did not appear in case of supercritical fluid extract due to the non-selectivity of the solvent extraction, which also impacted the color of the extract from white color to brown color due to relative extraction of other coloring matters and pigments in the fruits as shown in Figure 1.

The supercritical extraction method is a new separation technique that has been developed in recent years. Both temperature and pressure of the supercritical fluid are higher than the critical point, and thus the supercritical fluid has similar density to an ordinary fluid and many substances have a good solubility in it. Meanwhile, it also keeps the transfer properties and easy penetration characteristics of gas. Several studies have proved that SCF is advantageous over conventional solvent extraction. This may be due to the properties of supercritical fluids, which have different diffusivity, dielectric constant, density and viscosity. The high diffusivity, together with low viscosity of SCF, improve their diffusion through the plant material during the extraction process, especially under reduced pressure [34]. Altering the extraction conditions, such as extractor volume [34], temperature, pressure, and flow rate of both CO_2_ and the used co-solvent may also favor the extraction of certain compounds in the extract, which gives the SCFE technique a selectivity advantage over conventional extraction methods [35].

Moreover, SCFE has been reported to be superior to several conventional extraction methods, with regards to cleanliness, being environmental friendly, selectivity, time and cost saving in the long run, as well as possibility of manipulating the composition of the extracts according to the desired use [33].

*Ammi visnaga* extracts, as well as standard khellin and visnagin, were tested for their cytotoxic activity using SRB assay. Results revealed a strong cytotoxic activity (IC_50_ < 20 µg/mL) for the SCFE and standard compounds (khellin and visnagin) (IC_50_ ranging between 12.54 ± 0.57 and 17.53 ± 1.03 µg/mL). However, ethanol and acetone extracts had moderate cytotoxic activity (IC_50_ 20–90 µg/mL) and aqueous extract had a weak activity (IC_50_ > 90 µg/mL) [36]. Cytotoxic activity of *A. visnaga* extracts has been reported in several studies. Khellin and visnagin were also previously reported to have cytotoxic effects against several cell lines including MCF-7 and Hep G2 [37,38,39]. The high purity of furanochromones in SCFE may account for its better cytotoxic activity.

## 4. Materials and Methods

### 4.1. Plant Material

Fruits of *A. visnaga* were purchased from the Ministry of Agriculture & Land Reclamation, Giza, Egypt & planted in Mepaco medicinal farm. Plant identity was verified by Dr. Mohamed EL Gebaly, Department of Botany, National Research Centre in Egypt. Voucher specimens are deposited in the Pharmacognosy research lab at the Future University in Egypt and labeled 112-AV. Authentic khellin and visnagin were obtained from Sigma Aldrich (Darmstadt, Germany).

### 4.2. Preparation of the Plant Extracts

#### 4.2.1. Conventional Solvent Extraction

Maceration technique was applied for solvent extraction of total furanochromones from *A. visnaga* fruits using three different solvents. Different trials were used on a large scale using a solvent extraction system. Only solvents with minimum hazard were used; basically ethanol 95%, 30% ethanol, acetone and boiling purified water. Ten kg of *A. visnaga* fruits (previously crushed and sieved on sieve 1 mm) were macerated with ten times its weight of the recommended solvent with occasional stirring for 8 h using mechanical stirrer; then it was set aside overnight (12 h) at room temperature. The liquid extract was then filtered over thick cotton cloth. This step was repeated three times on the marc. The marc was then washed with double its weight with the recommended solvent. The filtrates were combined with the washing of the marcs, mixed well, and re-filtered using thick cotton cloth filter. The liquid extract was concentrated under vacuum at 5 °C until a final dry extract was obtained.

#### 4.2.2. Pilot Supercritical Fluid Carbon Dioxide Extraction Method

Ten kilograms of finely ground air-dried *A. visnaga* fruits (with particle size 1 mm) were packed in 20 L extraction vessel of 20 L ASI pilot scale SCFE unit (Applied Separations, Inc. Allentown, PA, USA), which is schematically presented in Figure 2. Extraction was carried out according to previously validated extraction conditions [40] The extraction was started with static time of 2 h, followed by dynamic extraction for another 2 h with flow rate of supercritical fluid carbon dioxide (containing 5% methanol as co-solvent) 2 L/min, supercritical pressure was 200 Bar and supercritical temperature of 45 °C.

### 4.3. HPLC Analysis

Determination of furanochromones in the extracts using standard calibration curves for khellin and visnagin was performed according to a validated method [41]. Analysis was performed using ultra-fast liquid chromatography: Shimadzu, Model Prominence LC-20ADXR, equipped with auto sampler, SIL-20ACXR), and PDA detector (Shimadzu Model: SPD-M20A), Kyoto, Japan. Column: ODS3 (250 × 4.6 mm 5 µm, 100 A, Phenomenex, Torrance, CA, USA). The injection volume was 20 µL, detection wavelength was 245 nm and mobile phase consisted of methanol: water (50:50 *v/v*) at flow rate 1.5 mL/min. *A. visnaga* extract (15 mg dissolved in methanol) were put in 20 mL volumetric flask and sonicated (concentration of 75 mg/dL). A 0.45 µm syringe filter was used for filtration of the solution. Standard khellin and visnagin (20 mg dissolved in methanol) were put in 100 mL volumetric flask, sonicated (concentration of 20 mg/dL) and filtered through 0.45 µm syringe filter prior to injection with same chromatographic conditions of the sample.

### 4.4. Cytotoxic Activity of A. visnaga Extracts, Standard Khellin, Visnagin and Doxorubocin

Two cell lines (MCF-7 and Hep G2) were used to test the cytotoxic activity of the extracts as well as pure khellin and visnagin. The cells were maintained in the laboratory of National Cancer Institute (Cairo, Egypt). Normal hamster lung fibroblasts (V79 cells) were used as control to assess the safety of samples on normal cells.

Sulforhodamine B stain assay was employed using concentrations 0–200 µg/mL [42]. Doxorubicin prepared in the previously mentioned concentrations was used as a reference. IC_50_ in µg/mL, as well as selectivity index (SI) were calculated.

### 4.5. Statistical Analysis

All performed analysis was carried out in triplicate. Values are expressed as mean ± SEM. Paired-*t*-test at *p* ≤ 0.05 was used to analyze significant difference using GraphPad Prism^®^ v.5.

## 5. Conclusions

Trials for conventional solvent extraction of furanochromones from *A. visnaga* fruits revealed that the highest yield of furanochromones was obtained using ethanol 95% (8.23% *w/w*). Using 30% ethanol was a better choice as a solvent, although it gave a slightly lower yield than acetone (6.59% *w/w*), but acetone is less hazardous, more economic and the extract had much better physical characters. Supercritical fluid extraction of furanochromones was the best method applied as it gave 30.1% *w/w* yield with excellent physical characters. Moreover, the SCFE had a stronger cytotoxic activity than the ethanol extract on MCF-7 and Hep G2 cell lines (IC_50_ 17.53 and 15.39 μg/mL, respectively). Therefore, SCFE is a relatively cheap, safe and environment friendly green technique that produced a more selective purified extract free from any residual solvent. This suggests that supercritical fluid extraction may be the method of choice for large scale extraction of furanochromones which are incorporated in several pharmaceutical products. However, trials for monitoring SCFE conditions may be carried out in future research to physically separate individual compounds (khellin and visnagin) in the same run. Also, comparison of different sources of *A. visnaga* fruits, as well as studying the effect of different cultivation conditions or plant organ on the yield of furanochromones using SCFE may be implicated in future research.

## Figures and Tables

**Figure 1 molecules-26-01290-f001:**
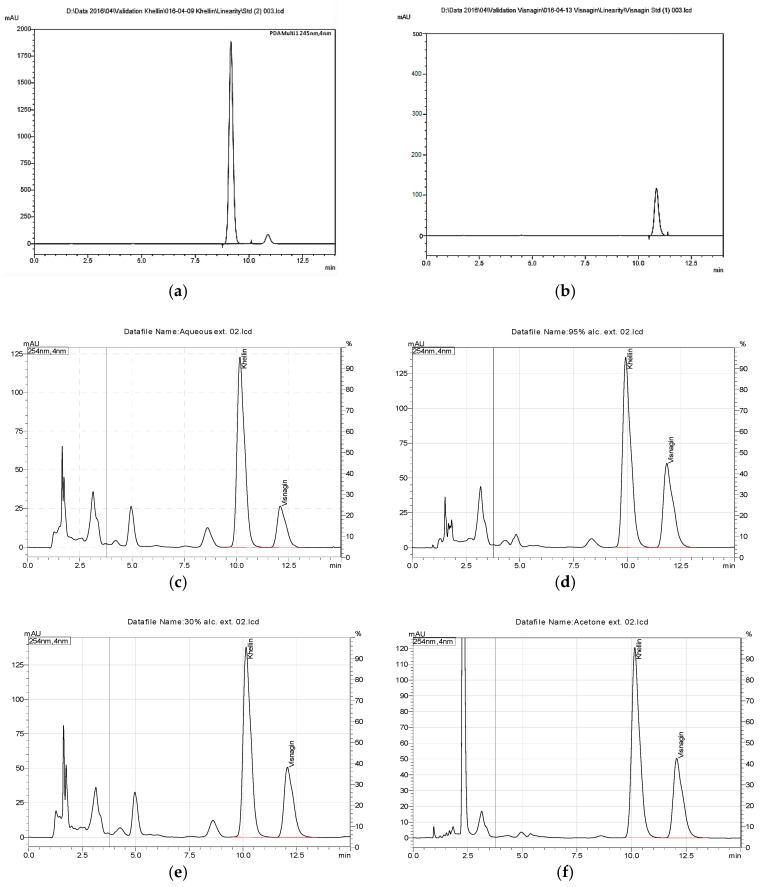
HPLC chromatograms of standard compounds and all tested extracts. (**a**) standard khellin, (**b**) standard visnagin, (**c**) aqueous extract, (**d**) 95% ethanolic extract, (**e**) 30% ethanolic extract, (**f**) acetone extract, (**g**) supercritical fluid extract.

**Figure 2 molecules-26-01290-f002:**
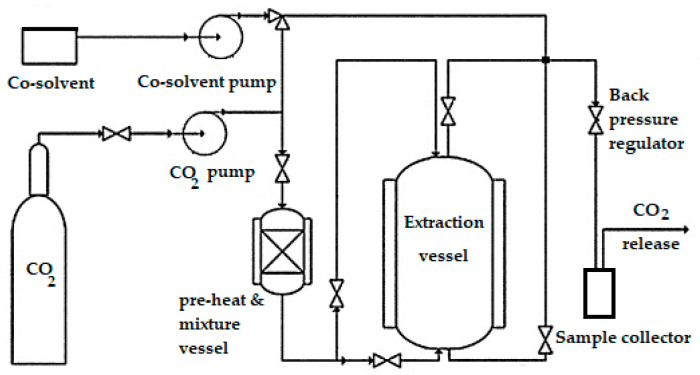
Scheme showing supercritical fluid extraction unit.

**Table 1 molecules-26-01290-t001:** Average yields and furanochromones content in *A. visnaga* tested extracts.

Solvent Used	Solvent Temperature (°C)	Physical Properties Of The Obtained Extract	Total Solid Extract Yield (G)	Total Solid Extract Yield (%)	HPLC Analysis			
					Kh. (%)	Vis. (%)	Kh. + Vis. (g)	Kh. + Vis.%
Boiling water	100	Soft brown sticky	1380.7 ± 30.22	13.807 ± 0.32	5.03 ± 0.06	0.92 ± 0.08	82.11 ± 2.33	5.95 ± 0.14
95% Ethanol	45–50	Soft light brown	1350.6 ± 25.67	13.506 ± 0.25	6.03 ± 0.07	2.20 ± 0.02	111.1 ± 3.58	8.23 ± 0.09
30% Ethanol	45–50	Soft dark brown	1544.1 ± 51.25	15.441 ± 0.51	5.03 ± 0.03	1.56 ± 0.02	101.74 ± 2.01	6.59 ± 0.05
Acetone	45–50	Soft fatty greenish	507.5 ± 11.69	5.075 ± 0.11	13.71 ± 0.12	4.78 ± 0.09	93.74 ± 3.45	18.49 ± 0.21
SCFE	45	Yellowish white, slightly fatty, strong characteristic odor	450.8 ± 8.58	4.508 ± 0.81	28.8 ± 1.22	2.01 ± 0.04	135 ± 3.22	30.1 ± 1.56

**Table 2 molecules-26-01290-t002:** In-vitro cytotoxic activity of *A. visnaga* extracts, standard khellin, visnagin and doxorubicin.

Test	Hep G2	SI	MCF-7	SI
	IC_50_ (μg/mL)		IC_50_ (μg/mL)	
Boiling water extract	112.58 ± 5.69 ^aB^	2.11	123.87 ± 6.22 ^aB^	2.69
30% Ethanol extract	51.22 ± 3.12 ^aB^	2.89	33.96 ± 2.45 ^aB^	3.11
95% Ethanol extract	45.28 ± 2.33 ^aB^	2.14	59.15 ± 4.25 ^aB^	2.55
Acetone extract	89.12 ± 4.23 ^aB^	1.23	88.14 ± 3.45 ^aB^	1.55
SCFE	17.53 ± 1.03 ^bB^	3.15	15.39 ± 0.58 ^bB^	4.57
Khellin	13.86 ± 0.88 ^bB^	3.96	12.54 ± 0.57 ^bB^	3.24
Visnagin	15.98 ± 0.66 ^bB^	3.78	13.79 ± 0.43 ^bB^	2.78
Doxorubicin	4.93 ± 0.26 ^A^		3.58 ± 0.31 ^A^	

Values are ± SEM (*n* = 3), means followed by different letters in same column denote significant difference at *p* < 0.05, paired-*t*-test. Lowercase letters compare means of test samples, uppercase letters compare means of sample with the standard doxorubicin. SI: Selectivity index was calculated as the ratio of the IC_50_ values on V79 cells to those in the tested cancer cell lines. SI > 3 indicates a promising activity.

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
