# Peer review of "Assessment of Conventional Solvent Extraction vs. Supercritical Fluid Extraction of Khella (Ammi visnaga L.) Furanochromones and Their Cytotoxicity"

_molecules, 2021, doi:10.3390/molecules26051290_

Round 1

Reviewer 1 Report

The paper entitled “Assessment of Conventional Solvent Extraction vs. Supercritical Fluid Extraction of Khella (Ammi visnaga L.) Furanochromones and their Cytotoxicity: A Comparison of Supercritical Fluid and Conventional Extraction Methods”  reports a study on the cytotoxicity effects and phytochemical content of extracts from Ammi visnaga leaves obtained. 

The article  can be improved if the following points are properly addressed:

The abstract needs to be revised. What is the indicator to show the cytotoxic is strong or moderate?

Please revise figure 1. It is not well organised. The standard calibration curves are not needed. 

Please show the HPLC chromatogram to prove that the SFE method selectively extracts the two compounds of interested and show the chromatogram of the conventional method as well. 

Please present the cytotoxic data in only table or figure but not both. 

Please revise the discussion part accordingly. It is good to state the beneficial of SFE, but it is too lengthy. Discussion is to discuss the significance of your result and finally the future prospect. 

The method of the conventional extraction method should be maceration as a whole.

Overall, the manuscript is quite simple and straight forward with a few experiments included. 

Author Response

Dear respected reviewer,

Thanks a lot for all your valuable comments. We hope we have responded to all your inquiries.

  • The abstract needs to be revised. What is the indicator to show the cytotoxic is strong or moderate?

Response: Ranges for cytotoxicity activity strength has been indicated in the abstract as directed (lines 22-25). The reference has been added in the discussion section (lines 143-146).

  • Please revise figure 1. It is not well organised. The standard calibration curves are not needed. 

Response: The figure has been organized and calibration curves have been removed as directed (lines 86-87).

  • Please show the HPLC chromatogram to prove that the SFE method selectively extracts the two compounds of interested and show the chromatogram of the conventional method as well. 

Response: All HPLC chromatograms are given in Figure 1 (lines 86-87).

  • Please present the cytotoxic data in only table or figure but not both. 

Response: the figure has been omitted as suggested.

  • Please revise the discussion part accordingly. It is good to state the beneficial of SFE, but it is too lengthy. Discussion is to discuss the significance of your result and finally the future prospect. 

Response: Lines 125-137 in the discussion section have been summarized as directed (This part was added upon the request of another reviewer).

  • The method of the conventional extraction method should be maceration as a whole.

Response: Maceration has been used for conventional extraction. (Do you mean we should remove the details of the maceration process?)

Reviewer 2 Report

The authors compared different extraction methods including supercritical fluid extraction to discuss the concentration of active components and its cytotoxicity. I have a few suggestions to improve the quality of this manuscript.

  1. In Table 1, the standard deviation of total solid extract yield (%) of SCFE is missing.
  2. Please revise Fig. 1. For example, scale of some subfigures are missing. Please also added illustration of all subfigures in figure caption.
  3. Water content of raw material is crucial in supercritical fluid extraction. If any pre-treatment was adopted for the raw material, please state in section 4.2.2

Author Response

Dear respected reviewer,

Thanks a lot for all your valuable comments. We hope we have responded to all your inquiries.

  • In Table 1, the standard deviation of total solid extract yield (%) of SCFE is missing.

Response: Missing SD have been added in Table 1 as directed.

  • Please revise Fig. 1. For example, scale of some subfigures are missing. Please also added illustration of all subfigures in figure caption

Response: Figure 1 has been corrected as directed (lines 86-87).

  • Water content of raw material is crucial in supercritical fluid extraction. If any pre-treatment was adopted for the raw material, please state in section 4.2.2

Response: The fruits were air-dried (added in line 174 as directed).

Reviewer 3 Report

The manuscript by Khalil and co-workers describes an scCO2-based extraction method for natural compounds, which is a timely and interesting topic. The work has some merits and the results are of interest to the readers of ‘molecules’. However, the major and minor issues outlined below needs to be addressed prior to further consideration.

1) The derivation of the reported errors should be included. Were independently prepared samples subjected to the extraction? How many samples were prepared and how many extractions were carried out that resulted in the errors. Are the errors standard deviations?

2) When reporting values, the accuracy of them should be taken into account. In many cases two decimal places are reported, which is not justified. The reported number of decimal places implies the accuracy of the method and obtained values and therefore should be carefully considered.

3) How general are the results obtained in this study? In other words, how much Khella fruits differ in terms of location, type, and time of processing?

4) The importance of green methods in natural product extraction should be emphasized and recent examples included before the introduction of the green scCO2 method (10.1021/acssuschemeng.9b04245; 10.1039/C7GC00912G; 10.1021/acssuschemeng.0c03393; 10.1021/acssuschemeng.0c08146).

5) Section 2.2 describes the HPLC method but it is very vague and not reproducible. Provide the detailed method to aid understanding and reproduction of your work.

6) Figure 3 should be more information-rich by including the process details such as concentrations, solvent, pressure, flow rate, temperatures etc.

7) The authors need to add some more critical evaluation of their work including the drawbacks and limitations of the proposed methodologies. The potential impact of the work and next steps should also be clearly communicated.

8) It is difficult to follow the article and find the information. The results and the discussion sections should be combined under a single section, and the Materials and methods section should be presented first, followed by the results and discussion. The authors should consider this restructuring.

9) The calibration curves on top of page 4 are unnecessary and should be deleted.

10) The bottom part of some of the spectra are missing. I do not know if this is a pdf conversion issue but this should be carefully revised during the resubmission as the results cannot be interpreted this way.

11) Both the quotient (“x/y”) and negative exponent (“x y-1”) formats are used in the manuscript for units. Either of them should be used consistently, preferably the negative exponent format, which is recommended by the IUPAC.

12) The conclusion section should have the main results summarized in quantitative statements as well. The authors should clarify how further research and applications will build on the research findings presented here.

13) Table 1 should have an additional column with the temperature for all solvents.

Author Response

Dear respected reviewer,

Thanks a lot for all your valuable comments. We hope we have responded to all your inquiries.

  • The derivation of the reported errors should be included. Were independently prepared samples subjected to the extraction? How many samples were prepared and how many extractions were carried out that resulted in the errors. Are the errors standard deviations?

Response: This has been mentioned in section 4.5 (lines 204-207).

  • When reporting values, the accuracy of them should be taken into account. In many cases two decimal places are reported, which is not justified. The reported number of decimal places implies the accuracy of the method and obtained values and therefore should be carefully considered.

Response: All values have been reported in two decimal places as directed.

  • How general are the results obtained in this study? In other words, how much Khella fruits differ in terms of location, type, and time of processing?

Response: Khella fruits used in this study were obtained from Egypt (as mentioned in section 4.1). However, it has been suggested for future research to compare the furanochromones yield from A. visnaga from different origins.

  • The importance of green methods in natural product extraction should be emphasized and recent examples included before the introduction of the green scCO2 method (10.1021/acssuschemeng.9b04245; 10.1039/C7GC00912G; 10.1021/acssuschemeng.0c03393; 10.1021/acssuschemeng.0c08146).

Response: Examples of green methods using the suggested references have been added in the Introduction section as directed (lines 46-50).

  • Section 2.2 describes the HPLC method but it is very vague and not reproducible. Provide the detailed method to aid understanding and reproduction of your work.

Response: Details for HPLC method has been mentioned in details in section 4.3 (lines 183-195).

  • Figure 3 should be more information-rich by including the process details such as concentrations, solvent, pressure, flow rate, temperatures etc.

Response: The figure was added upon the request of another reviewer to show a schematic diagram of an SCF extraction unit. All details are mentioned in section 4.2.2. Do you mean to add the details on the scheme?

  • The authors need to add some more critical evaluation of their work including the drawbacks and limitations of the proposed methodologies. The potential impact of the work and next steps should also be clearly communicated.

Response: Potential impact of the work and future research suggestions have been added in conclusion section as directed (lines 218-225).

  • It is difficult to follow the article and find the information. The results and the discussion sections should be combined under a single section, and the Materials and methods section should be presented first, followed by the results and discussion. The authors should consider this restructuring.

Response: The article was structured according to the journal guidelines.

  • The calibration curves on top of page 4 are unnecessary and should be deleted.

Response: The calibration curves have been removed as directed.

  • The bottom part of some of the spectra are missing. I do not know if this is a pdf conversion issue but this should be carefully revised during the resubmission as the results cannot be interpreted this way.

Response: Figure 1 has been clarified as directed (lines 86-87).

  • Both the quotient (“x/y”) and negative exponent (“x y-1”) formats are used in the manuscript for units. Either of them should be used consistently, preferably the negative exponent format, which is recommended by the IUPAC.

Response: All quotients are used as x/y throughout the whole manuscript.

  • The conclusion section should have the main results summarized in quantitative statements as well. The authors should clarify how further research and applications will build on the research findings presented here.

Response: Missing quantitative statements have been added in conclusion section as directed. Also, potential impact of the work and future research suggestions has been added in conclusion section as directed (lines 218-225).

  • Table 1 should have an additional column with the temperature for all solvents.

Response: Solvents temperatures have been added in Table 1 as directed.

Round 2

Reviewer 1 Report

Dear author, 

Thank you for the correction. 

In section 4.3 HPLC part, you did mention the standard preparation. However, in figure 1, the chromatogram for the standard is missing. We do find a clean chromatogram label as SFE extract. please clarify. 

Author Response

Dear respected reviewer,

Thanks a lot for your comment. Chromatograms for standard compounds have been added to figure 1 as directed. The manuscript has also been spell-checked as required.

Thank you.

Reviewer 3 Report

The authors have addressed the comments and the manuscript has improved. In this reviewer's opinion, the manuscript can be accepted for publication in Molecules.

Author Response

Dear respected reviewer,

The manuscript has been spell-checked as required.

Thank you.

This manuscript is a resubmission of an earlier submission. The following is a list of the peer review reports and author responses from that submission.

Round 1

Reviewer 1 Report

The manuscript compared the conventional extraction yield and the cytotoxicity activity of the extracts of Ammi Visnaga. 

Major concern

Figure 1 showed a comparison of 30% ethanol extract and SFE extract. The figure didn't show any relative improvement of the extraction method as the margin is incorrect. Where is the chromatogram of the standard compounds? 

The % of furanochromones in SFE is rather outrageous higher than other extraction methods. The authors should show the chromatogram of each extract and standardise it. 

Most of the SFE extraction method yield is lower than the conventional method in regards to the conventional extraction method is not selectively extract more compounds.  

Why only compare ethanol and SFE but not all non-conventional method with SFE for the cytotoxicity assays? 

Are 200 bar, 45C and 5% of ethanol the optimum extraction method to extract the bioactive compounds? Any preliminary study for it before proceeding with the pilot-scale extraction method. 

Reviewer 2 Report

Comment

  1. In this manuscript, SCFE is considered as a “cheap” technique. However, compared with the conventional extraction process, in general, the cost of SCFE is still high. To claim SCFE is a cheap technique, the authors should provide some evidence of economic evaluation.
  2. In Fig. 1, the time scale of two chromatograms should be identical. In addition, the HPLC chromatograms of two standards should be included in Fig. 1 for comparison clearly.
  3. In Table 1, discussion of results of doxorubicin is missing in this manuscript.
  4. The authors used 30% ethanol as the extraction medium to replace the expensive ethanol. If possible, please add results for the extraction experiment using ethanol as the solvent for comparison.
  5. In section 4.2, a detailed descriptions of experimental conditions are required. For example, the extraction temperature of conventional solvent extraction and how to decide the extraction conditions of SCFE.
  6. A figure to graphically present the experimental system of SCFE should be included in Section 4
  7. In results and discussion section, the authors should discuss the effect of process parameter in SCFE and explained in theory. Now, the manuscript was more like a report.